# Mitigation of Soil Erosion and Enhancement of Slope Stability through the Utilization of Lignin Biopolymer

**DOI:** 10.3390/polym16091300

**Published:** 2024-05-06

**Authors:** Pouyan Bagheri, Ivan Gratchev, Masih Zolghadr, Suwon Son, Jin Man Kim

**Affiliations:** 1School of Engineering and Built Environment, Griffith University, Engineering Drive, Southport, QLD 4222, Australia; 2Department of Water Sciences and Engineering, Jahrom University, Jahrom 74148-46199, Iran; 3Department of Architectural and Civil Engineering, Kyungil University, Gyeongsan 38428, Republic of Korea; 4Department of Civil and Environmental Engineering, Pusan National University, Busan 46241, Republic of Korea

**Keywords:** lignin biopolymer, soil erosion, soil strength, flume experiments

## Abstract

Human activities have had a profound impact on the environment, particularly in relation to surface erosion and landslides. These processes, which are natural phenomena, have been exacerbated by human actions, leading to detrimental consequences for ecosystems, communities, and the overall health of the planet. The use of lignin (LIG) as a biopolymer soil additive material is regarded as an eco-friendly solution against soil erosion and slope failure which holds immense promise. However, significant research gaps currently hinder a comprehensive understanding of its mechanisms and effectiveness. Experimental studies offer a robust platform to address these gaps by providing controlled conditions for assessing soil stability, exploring mechanisms, and evaluating adaptability. Bridging these research gaps will contribute to the development of innovative and sustainable strategies for mitigating soil erosion and preventing slope failure, thereby promoting environmental resilience and resource conservation. This study aimed to investigate the effect of the LIG biopolymer on mitigation of soil erosion, slope failure and the enhancement of soil strength by conducting laboratory tests (UU triaxial, unconfined compressive strength (UCS), and soaking) as well as flume experiments under uniform rainfall events. The alterations in the engineering characteristics and erosion resistance of silty soil mixed with a LIG additive at concentrations of 1% and 3.0% by weight have been examined. The results show that the LIG-treated samples demonstrated an enhanced resistance to surface erosion and an enhanced prevention of slope failure, as well as improved shear stress, cohesion, stiffness, and resistance to water infiltration.

## 1. Introduction

Soil erosion, slope failure, and landslides result from the combined effects of rainfall, topography, land use, vegetation cover, and human activities. These processes can lead to the degradation of agricultural lands, loss of fertile topsoil, degradation of land productivity, increased sedimentation in water bodies, destruction of infrastructure, sedimentation of water bodies, and even loss of lives. Therefore, preventing soil erosion and enhancing slope stability by various physical and chemical approaches has long been a popular research area on a global scale.

A slope protection technique using eco-friendly materials is a sustainable method that improves a slope’s ability to resist erosion. In recent years, biopolymers have emerged as a promising alternative owing to their eco-friendly nature, biodegradability, and potential to improve soil properties [1]. Biopolymers are naturally occurring polymers produced by living organisms, such as plants, bacteria, and animals. They are biodegradable and possess properties that make them attractive for various geotechnical engineering applications, including soil and slope stabilization.

Recent studies have demonstrated the usage of biopolymers in geotechnical engineering applications. Guar gum [2], lignin [3], agar [4,5], beta-glucan [6], and alginate [7] are instances of plant-based biopolymers whose uses in geotechnical engineering have already been investigated. Jang et al. [5] reported that the liquefaction resistance strength of soil increased with higher concentrations of agar gum. Chitosan, the deacetylated form of chitin that is obtained from the wastes of marine food production, and casein, a protein obtained via the acidification or enzymatic action of milk and/or dairy products, are animal-originated examples of biopolymers whose applications in geotechnical engineering have been researched [8,9]. Hataf et al. [9] demonstrated that under wet conditions, chitosan can improve the interparticle bonding among soil particles. Microorganism-based biopolymers such as xanthan gum [1] and gellan gum [10] have shown their applications in soil properties improvement. Bagheri et al. [1] investigated the impact of xanthan gum on soil strength and verified a significant enhancement in soil compressive strength over a specific curing period. 

The effects of xanthan gum, beta glucans, guar gum, chitosan, and alginate biopolymers on the strength of silty sand soil were studied by Soldo et al. [7]. They confirmed a considerable soil strength enhancement over a period of time. Ham et al. [11] investigated the impact of dextran, a microbial biopolymer, on surface erosion. The effect of gellan gum biopolymer’s on soil permeability was examined by Chang et al. [12]. 

Lignin is a complex and heterogeneous biopolymer that is a major component of plant cell walls, particularly in woody tissues. It is the second most abundant organic material on earth after cellulose. Lignin provides structural support to plants, contributing to their mechanical strength and rigidity. Traditionally, lignin has been considered a waste product of the pulp and paper industry. However, recent research has revealed its potential in various applications, including geotechnical engineering practices as reported by various researchers [3,13,14]. Bagheri et al. 2023 [3] conducted a series of laboratory experiments and reported that adding a lignin biopolymer resulted in higher soil strength. Yang et al. [15] investigated the effect of lignin addition on silty soil shrinkage performance and reported that the shrinkage characteristics of lignin-treated silt were influenced by the combined function of lignin cementation and matric suction.

As mentioned, in recent years, the need for sustainable and eco-friendly solutions to mitigate soil erosion and slope failure has led to a growing interest in exploring novel soil stabilization methods. One such emerging solution involves the use of lignin-treated soil, which shows promise due to its environmentally friendly properties and potential to enhance soil stability. One of the significant gaps in the current literature is a comprehensive understanding of how lignin-treated soil interacts with the soil matrix. While the adhesive properties of lignin are well-established, its molecular-level interactions with soil particles and its effects on soil structure need further exploration. This paper highlights the research gaps in understanding the mechanisms and effectiveness of lignin-treated soil in combating soil erosion and slope failure and provides a rationale for conducting laboratory flume experiments to bridge these gaps. Additionally, few studies have delved into the effects of biopolymer treatment on soil strength under real-world, three-dimensional field conditions. As a result, a comprehensive set of laboratory and flume tests was carried out to evaluate the eco-friendly, lignin-based treatment’s efficacy in mitigating soil erosion, preventing slope failure, and bolstering soil strength. 

## 2. Materials and Methods

A silt soil with low plasticity (classified as ML according to (ASTM D2487-17) [16]) was acquired from the Gold Coast region in Australia. The grain size distribution depicted in Figure 1 was determined using (ASTM D422-63) [17] as a guideline. Atterberg limits testing (ASTM D4318-17) [18] and a specific gravity test in accordance with (ASTM D854-14) [19] were performed to obtain soil properties. Table 1 summarizes the soil properties.

A standard proctor compaction test (ASTM D698-12) [20] was conducted, and a maximum dry density of 1.72 g/cm^3^ and corresponding optimum water content of 21.7% were attained.

The ammonium lignosulfonate used as lignin (referred to as LIG) was sourced from Dustex in Australia. This material was in the form of a brown, thick liquid and had a pH of 5.4 ± 3.0 when in a 10% solution. The LIG consisted of a combination of water (51%) and ammonium lignosulfonate (49%).

### 2.1. Sample Preparation

For the first part of the research, the soil was initially dried in an oven, and then the gravel was separated by being crushed and filtered through a sieve with a 2.36 mm opening. The study utilized two different proportions of dry soil mass to LIG (1.0 wt.% and 3.0 wt.%). The selection of these concentrations was driven by two primary factors. Firstly, challenges were encountered when using higher LIG concentrations exceeding 3%, particularly in terms of workability during soil mixing, making them impractical for the experimental setup. Additionally, preliminary experiments suggested that excessively low LIG concentrations might not achieve significant effectiveness compared to higher doses. Therefore, in order to strike a balance between workability and efficacy, the concentrations of 1% and 3% LIG were chosen for further research investigation. For creating the LIG soil mixture, the wet mixing method, outlined by Ta’negonbadi and Noorzad [13], was utilized. This involved adding LIG liquid to water to attain the desired moisture level. The diluted solution was then sprayed onto the dry soil and thoroughly combined to achieve a uniform blend. The resulting mixtures were encased in double layer plastic wrap and stored in a temperature-controlled room for 24 h. This step was taken to prevent clumping and to ensure even integration of the biopolymer with soil particles. These mixtures were then packed into a cylindrical metal mold with a diameter of 50 mm and length of 150 mm (Figure 2a) and compacted evenly into 5 layers to form the samples. After each compaction cycle, samples with a 50 mm diameter and a length approximately measuring 110 mm were extracted from the mold (Figure 2). It was ensured that the dry density of each sample surpassed 95% of the maximum dry soil density.

### 2.2. Laboratory Experiments

To define the effect of the LIG biopolymer on the soil compressive strength, UCS tests for the untreated samples and LIG-treated specimens were performed. The UCS test results were incorporated from our prior study [3], as the current research serves as an extension of our previous investigation. It is important to highlight this continuity in our research efforts. The prepared specimens described in the previous section were placed in a temperature-controlled environment for curing, spanning durations of 0, 1, 7, 10, 14, 28, and 35 days. Subsequently, the cured specimens underwent UCS tests, following the methodology outlined in reference (ASTM D2166-06) [21]. It is worth noting that three specimens were subjected to each test to reduce potential inaccuracies. Based on the results obtained from the UCS tests, the ideal curing duration for both the treated and untreated specimens was identified and selected for subsequent experiments. 

To mimic the soil stress circumstances that exist in the field and obtain soil shear parameters, Unconsolidated Undrained (UU) triaxial tests on the LIG-treated and untreated samples were performed. The specimens were subjected to three different confining pressures (50, 100, and 200 kPa). The process of preparing the samples aligned with the conditions set for the UCS tests. No saturating procedure was used, and the dry specimen was initially exposed to the specified confining pressure, followed by the direct application of shear stress.

Soaking tests on the LIG-treated and untreated samples were carried out to identify when water caused the specimens to disintegrate. After being submerged in water for one day, all prepared specimens were evaluated visually for the degree of disintegration.

### 2.3. Flume Experiments

To examine soil–LIG interaction and how LIG contributes to soil cohesion and resistance against erosive forces, a series of laboratory flume experiments were conducted. This study utilized a custom-designed rainfall erosion test simulator, and Figure 3 illustrates the schematic diagram of this simulator. 

The flume box used for the tests is illustrated in Figure 3 with dimensions 600 mm long, 400 mm wide, and 100 mm high. The lower section of the flume box was built in a wedge shape with an angle of 45°. This allows free movement of soil particles and a possible sliding of soil inside the box. Two flume tests for treated soil with 1% and 3% LIG and one flume experiment for untreated soil were performed. In the case of treated flume experiments, the dried soil was first mixed uniformly with the given diluted LIG solution followed by a similar procedure detailed in Section 2.1. For the untreated soil flume test, a similar approach was sought by adding only water to the dry mass of soil. The soil mixture was then evenly distributed and compacted to a dry density of 1.1 g/cm^3^ within three equal layers. The box was mounted on a table and fixed to a slope of 30° gradient to achieve a slope resembling field conditions. The flume box was then covered with plastic wrap for 24 h curing. Each treated and untreated soil mixture box was subjected to four consecutive rainfall events. A rainfall intensity of 100 mm/h within a duration of 90 min for each event was adopted. A flowmeter and a valve regulator were attached to hoes connected to the sprinklers to ensure uniform rainfall intensity. The intensity of each rainfall event was measured by rain gauges. The eroded soil and flowing water were collected at various time intervals to obtain the rate of soil loss due to surface erosion and stream (runoff). The collection for each event was at various intervals (5, 10, 15, 30, 45, 60, 75, and 90 min) from the commencement of the experiment. The surface erosion and ground movement causing slope failure were assessed based on the flume box experiment results. 

## 3. Results and Discussions

### 3.1. UCS Tests

The variations in UCS for the untreated and LIG-treated soils across different curing durations are shown in Figure 4. 

Higher levels of LIG concentration led to increased UCS. Although the specimen treated with 1% LIG achieved its peak strength after a curing period of 10 days, the specimen treated with 3% LIG reached its maximum strength after curing for 14 days. The UCS significantly improved within a specific curing duration for all LIG concentrations, indicating that extended curing time had a minor influence on soil strength. While the specimen treated with 3% LIG exhibited nearly 2.5 times higher UCS than the untreated sample, incorporating 1% LIG into the soil resulted in a doubling of the soil’s compressive strength within the optimum curing time. As the treated samples dried, the LIG biopolymer functioned as a binding agent, resulting in a noticeable increase in soil strength. The curing duration associated with the highest UCS was determined as the optimum and was utilized for subsequent tests.

The effect of the LIG biopolymer on soil stiffness was assessed by secant stiffness (*E*_50_), of specimens from UCS tests. The graph in Figure 5 illustrates the secant stiffness values for specimens treated with LIG and the untreated soil samples. Incorporating LIG into the soil appears to enhance its stiffness. Both treated and untreated soil samples exhibited increased stiffness as the curing time progressed, with little change observed after 10 days.

Similar studies prove the positive effect of polymer stabilizers on soil properties. In a study [22], UCS and curing time of the soil were enhanced so that the studied polymer (vinyl acetate-ethylene) stabilized the clay samples through a pore-filling effect, physicochemical bonds, and surface wrapping. Kocak and Grant [23] implemented a commercially available liquid polymer instead of ordinary cement to assess soil stabilization during rammed earth construction. They reported that the CBR and UCS values of polymer-stabilized soils were improved up to 10 and 3 times, respectively, compared to untreated samples. Considering the environmentally friendly aspects of biopolymers, this study focuses on this type of polymer. 

### 3.2. UU Triaxial Tests

Figure 6 displays the deviatoric stress–axial strain curves obtained from UU triaxial tests. The addition of the biopolymer is observed to enhance soil strength, with a higher percentage of LIG resulting in more significant improvements. Regardless of LIG content, all specimens exhibited increased strength under higher confining pressures. Specimens treated with just 1% LIG experienced a notable 92% increase in maximum deviatoric stresses at 50 kPa confining pressure, highlighting the significant enhancement achievable with the minimal biopolymer concentration. Upon drying, LIG-treated soil transformed into a sturdy, stiff material, fostering strong inter-particle bonding and increased resistance to shearing. 

Prior research from direct shear tests has indicated that changes in shear parameters depend on the specific biopolymer and soil type; Cho and Chang [24] conducted a study to examine the impact of gellan gum on soil cohesion and friction angle. They employed a mixture of sand and clay and found that the addition of gellan gum enhanced the cohesion of pure sand. While the friction angle of pure sand remained relatively consistent as the concentration of gellan gum increased. The addition of gellan gum to pure clay resulted in both increased cohesion and an increased friction angle. In a separate study, Khatami and O’Kelly [25] reported that although the addition of agar and starch biopolymers to sand effectively increased soil cohesion, it simultaneously led to a reduction in the soil’s friction angle when compared to untreated soil. Ayeldeen et al. [26] conducted direct shear tests and demonstrated that xanthan gum and guar gum significantly increased soil cohesion while slightly decreasing the soil’s friction angle. Soldo et al. [7] investigated the shear behavior of silty sand treated with xanthan gum, guar gum, and beta-glucan biopolymers through direct shear tests. Similarly, their findings indicated that the addition of biopolymers increased soil cohesion but resulted in a reduction in the soil’s friction angle. Bagheri et al. [1] reported that for the dried samples adding a xanthan gum (XG) biopolymer improved soil strength and cohesion. Moreover, the samples treated with XG showed remarkable resistance against the decline in compressive strength resulting from repeated wetting and drying cycles.

Various studies have shown that biopolymers can increase soil cohesion but may reduce soil friction angles. In this study, UU triaxial test results were used to derive shear parameters and failure envelope curves were plotted (Figure 7), yielding corresponding shear parameters (Table 2). The addition of LIG resulted in a substantial boost in soil cohesion; just 1% LIG doubled the soil cohesion. This effect is attributed to LIG acting as an adhesive, providing robust binding between soil particles and consequently elevating soil cohesion. Nonetheless, there are no significant differences in soil cohesion enhancement among the given 1% and 3% LIG-treated soil. The sample treated with 1% LIG showed a marginal decrease in the soil friction angle, while the specimen treated with 3% LIG displayed a minor increase, suggesting that the LIG biopolymer had a negligible impact on the soil’s friction angle. 

### 3.3. Soaking Test

Figure 8 illustrates the results of a water immersion test conducted on untreated and LIG-treated specimens for a duration of 1 day. Upon immersion, the untreated soil specimen rapidly began to disintegrate and fully disintegrated within 4 h. In contrast, the LIG-treated specimens exhibited a notable difference. The LIG-treated specimens retained their shapes during the initial 8 h period but began to disintegrate thereafter. These specimens collapsed the following day after 24 h of immersion in water.

### 3.4. Flume Experiments

Soil erosion and its potential consequence, slope failure, were investigated in untreated and LIG-treated soil by rainfall-induced flume experiments. The rate of eroded soil mass and surface runoff for the first and second events of the untreated soil are shown in Figure 9. The erosion rate is defined as eroded soil mass over each time step of the experiment. Similarly, the runoff rate was determined by dividing the amount of flowing water over each time step. While there was an immediate spike in surface erosion for the first 5 min after starting the test, the erosion rate substantially decreased within the first 30 min of the experiment and stabilized afterward (Figure 9a). On the other hand, surface runoff significantly increased within the first 10 min of the experiment and, after a sharp decrease for the second 10 min of the test, stabilized to a rate of 350 (gr/min), (Figure 9b). A potential slope failure during each rainfall event was visually assessed. In the first event, no sign of a potential slope failure including cracks or ground movement was witnessed. 

The soil loss for the second rainfall event began with a slight rise for the first 10 min and the erosion rate flattened within the second 10 min. After that, the surface erosion drastically increased, and a marked soil loss is seen for the whole duration of the experiment (Figure 9a). An almost similar trend for the surface runoff is also seen for the second rainfall event, indicating soil infiltration and consequently surface runoff stabilized after the first 30 min of the test (Figure 9b). 

Due to continuous rainfall, the slope began to lose stability 20 minutes into the second event, leading to a rotational landslide (Figure 10b). This corresponds to the beginning of major soil loss presented in Figure 9a. The failure progress proceeded afterward (Figure 10c). At the end of the second experiment, a mass movement of soil was observed, and the slope completely collapsed indicating a clear rotational landslide due to continuous rainfall (Figure 10d).

Due to the occurrence of slope slip failure, the following rainfall events (third and fourth) for the untreated flume box were not conducted. 

The soil erosion rate and surface runoff for the four rainfall events for the LIG-treated soil are shown in Figure 11. For the first rainfall event, a similar pattern for both concentrations of LIG soil mixtures is seen. Over the first 5 min of the experiment, there is a significant rise in erosion rate followed by a sharp drop for the following 15 min of the experiment. Over the next 10 min, both treated soils reached their peak followed by a remarkable drop in erosion rate for the next 15 min lasting until 45 min after starting the experiment, and eventually, the soil erosion stabilized and merged to a fixed value rate. Throughout the experiment, the soil treated with 1% LIG showed a lower mass of eroded soil. A similar trend for the surface runoff in both treated soils is seen. During the initial 10 min of the experiment, there was a notable rise in surface runoff. Subsequently, there was a sudden drop in the runoff during the following 10 min of the test. However, it then leveled off at a consistent rate.

Contrary to the first event, over the following rainfall events, both LIG-treated soils exhibited a sharp increase in erosion rate for the first 10 min of the experiment. This was then followed by a sudden decrease in eroded soil mass for the next 10 min. A similar pattern to the first event is seen for the following section of the experiment with the erosion rate peaking at 30 min of the experiment and then stabilizing to a constant rate of soil loss. For all surface runoff rates, both concentrations of LIG-treated soils experienced a significant increase in surface flow water within the first 10 min and then a sudden decrease for the following 10 min, followed by a rise in the next 10 min lasting for 30 min of each event. The next hour of the experiment was followed by a slight decrease in flowing water for about 10 min and this stabilized for the following section of the experiment. 

The progress of the flume experiments for both treated soils during all rainfall events demonstrated that despite the obvious occurrence of surface erosion over all the experiments, no sign of any slope failures including sliding was observed. This proves the applicability of the LIG biopolymer in improving the attachment of soil particles and confirming the usage of such an eco-friendly technique against potential slip-prone areas. Figure 12 illustrates the schematic process by which soil properties are enhanced against surface erosion and slope failure. 

While this study presents promising results, there are inherent limitations to laboratory experiments that may impact the generalizability of the findings to real-world field scenarios. Laboratory experiments provide controlled conditions that allow for the rigorous testing and evaluation of specific variables. However, these conditions may not fully replicate the complex and dynamic environment found in the field. Therefore, despite the encouraging outcomes observed in our study, it is imperative to verify these results through large-scale field experiments.

## 4. Conclusions

This study examines the effect of different concentrations of LIG as a biopolymer soil additive on the enhancement of soil strength and soil slope stability and the resistance to erosion through laboratory experiments. The following provides a concise overview of the most noteworthy discoveries.

LIG-treated soil has been shown to be an effective method for mitigating soil erosion and slope failure.

The addition of LIG to soil increases its shear strength, cohesion, and stiffness. The soil’s compressive strength and cohesion doubled upon the introduction of 1% LIG. This is due to the adhesive properties of LIG, which binds soil particles together more tightly.

A moisture susceptibility test demonstrates LIG’s ability to uphold soil stability and improve its resistance to water infiltration. This is because the LIG molecules form a barrier between the soil particles and the water, preventing the water from penetrating the soil. This resistance to water infiltration can help to prevent slope failures by reducing the amount of water that can seep into the soil and weaken its structure.

In the laboratory flume experiments, untreated soil exhibited substantial soil erosion and a pronounced slope failure after the second rainfall event. In contrast, soil treated with LIG demonstrated slope stability even after enduring four consecutive extreme rainfall events. These findings indicate that the application of LIG treatment holds promise as an effective method for mitigating soil erosion and preventing slope failure.

The outcomes from the flume experiments revealed the resilience of LIG-treated soil against multiple rainfall events, demonstrating its ability to withstand such conditions without encountering slope failures. This significance implies that LIG-treated soil may offer a viable solution for areas prone to slope failure.

Despite above-mentioned promising results, some points need to be considered for the practical application of such a technique, including: 

The effectiveness of LIG-treated soil may vary depending on the type of soil, the concentration of LIG, and the environmental conditions.

More research is needed to fully understand the mechanisms by which LIG-treated soil enhances soil stability and to optimize the use of LIG-treated soil in different applications.

## Figures and Tables

**Figure 1 polymers-16-01300-f001:**
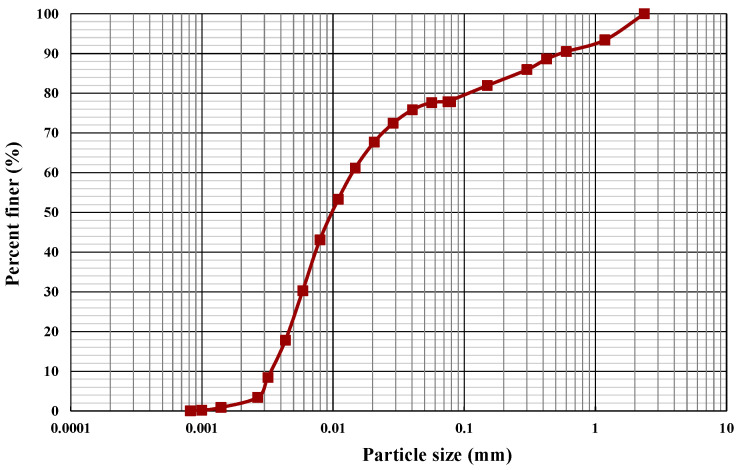
Grain size distribution.

**Figure 2 polymers-16-01300-f002:**
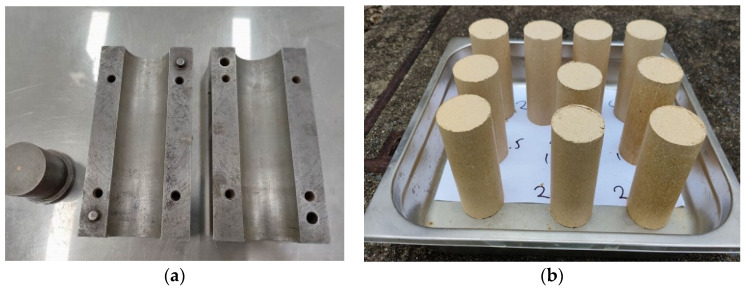
(**a**) Cylindrical metal mold and (**b**) prepared specimens.

**Figure 3 polymers-16-01300-f003:**
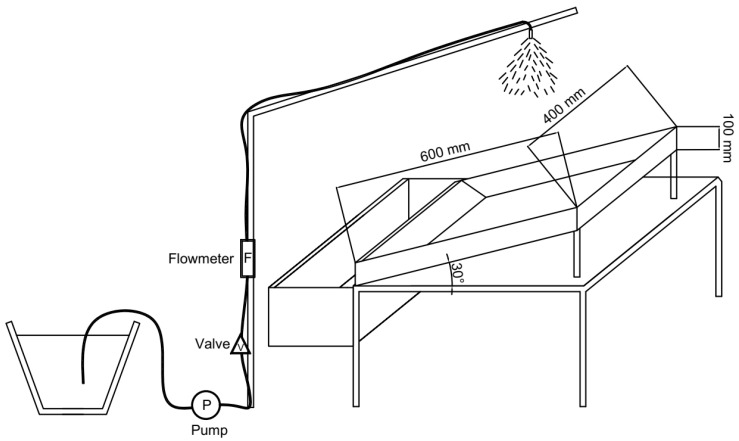
Schematic diagram of rainfall simulator.

**Figure 4 polymers-16-01300-f004:**
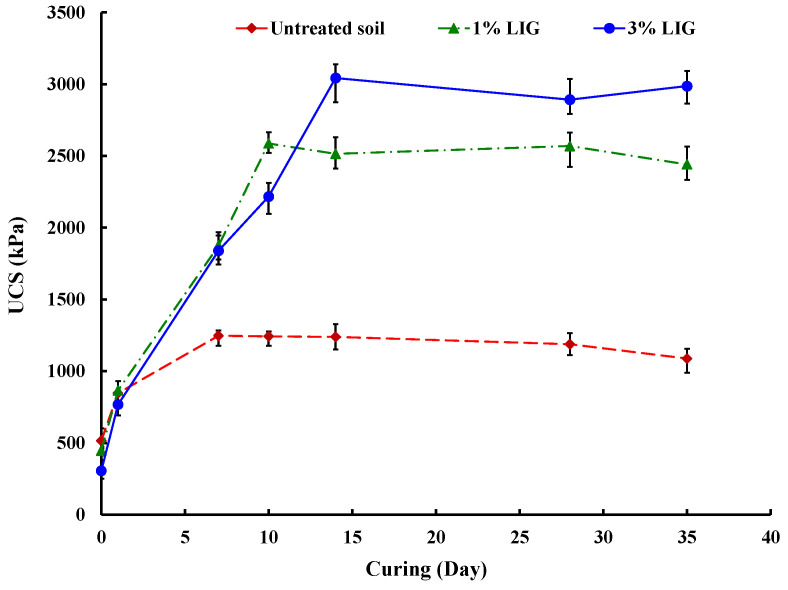
UCS changes with time for untreated and LIG-treated soil [3].

**Figure 5 polymers-16-01300-f005:**
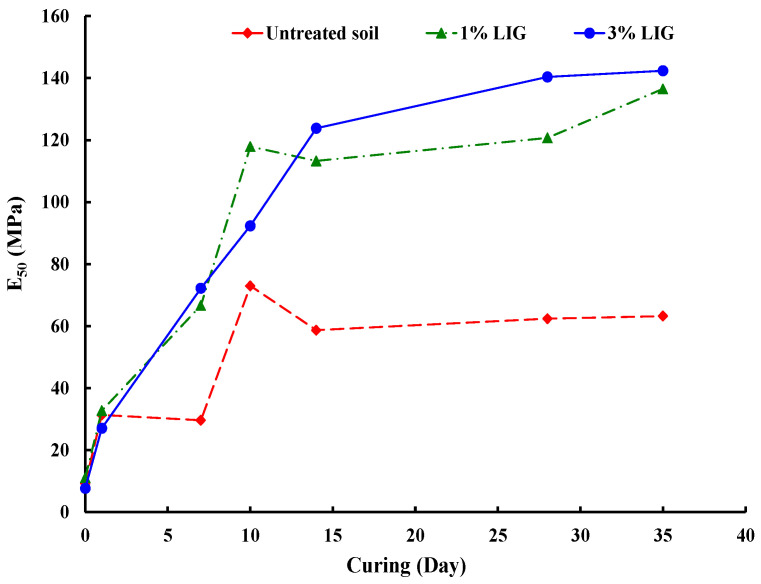
The correlation between curing time and secant stiffness across various LIG biopolymer content levels.

**Figure 6 polymers-16-01300-f006:**
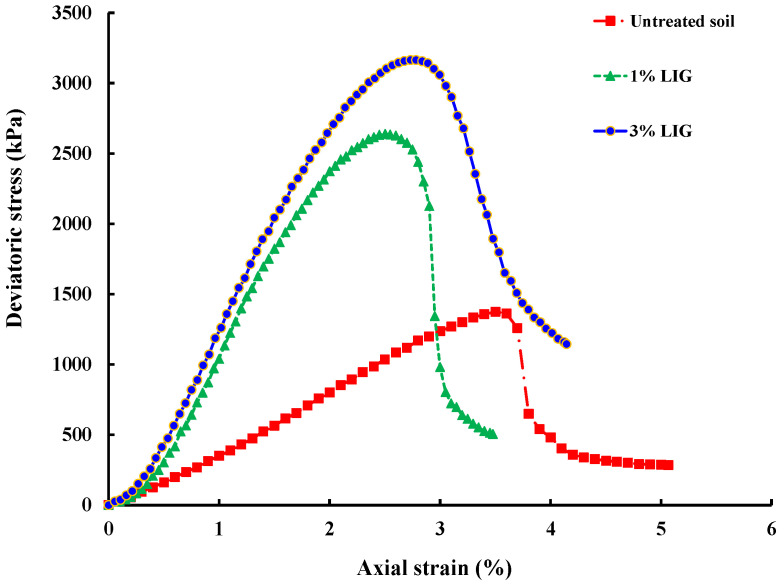
Stress–strain curves obtained from UU triaxial tests at 50 kPa confining pressure.

**Figure 7 polymers-16-01300-f007:**
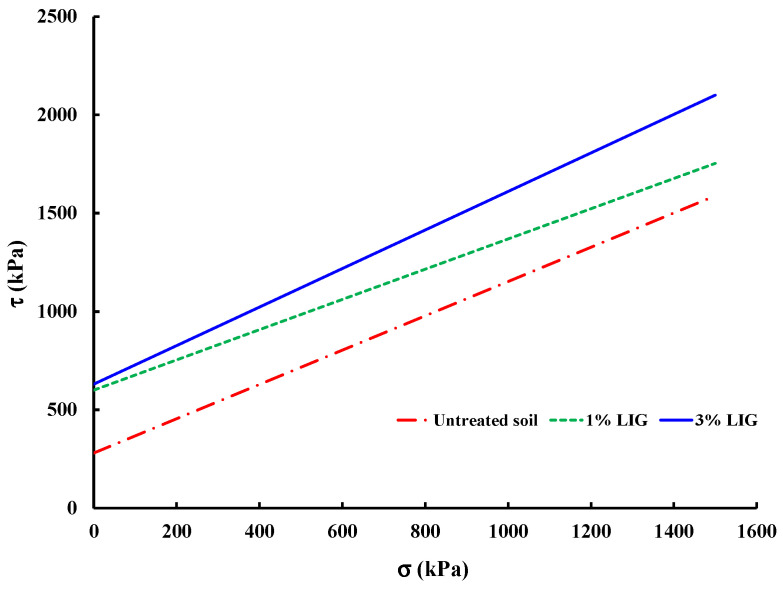
Plotted failure envelope curves obtained from UU triaxial tests.

**Figure 8 polymers-16-01300-f008:**
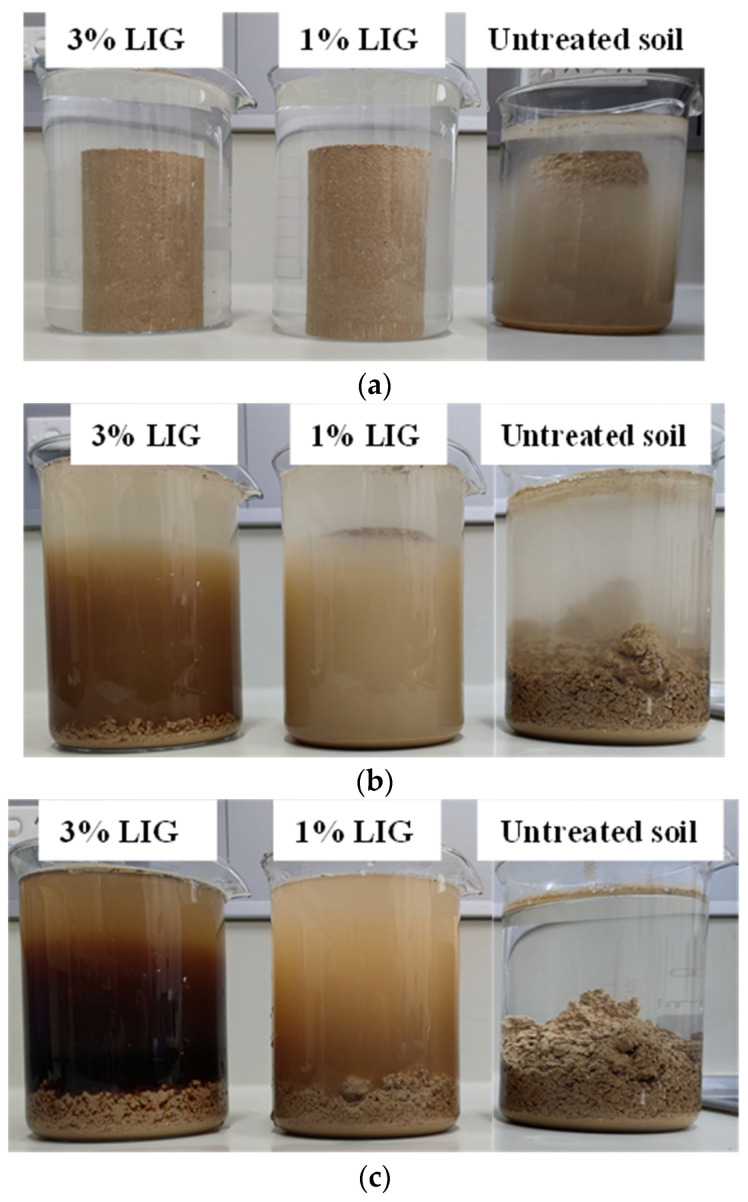
Specimens immersed in the water (**a**) after 5 min, (**b**) after 4 h, and (**c**) after 24 h.

**Figure 9 polymers-16-01300-f009:**
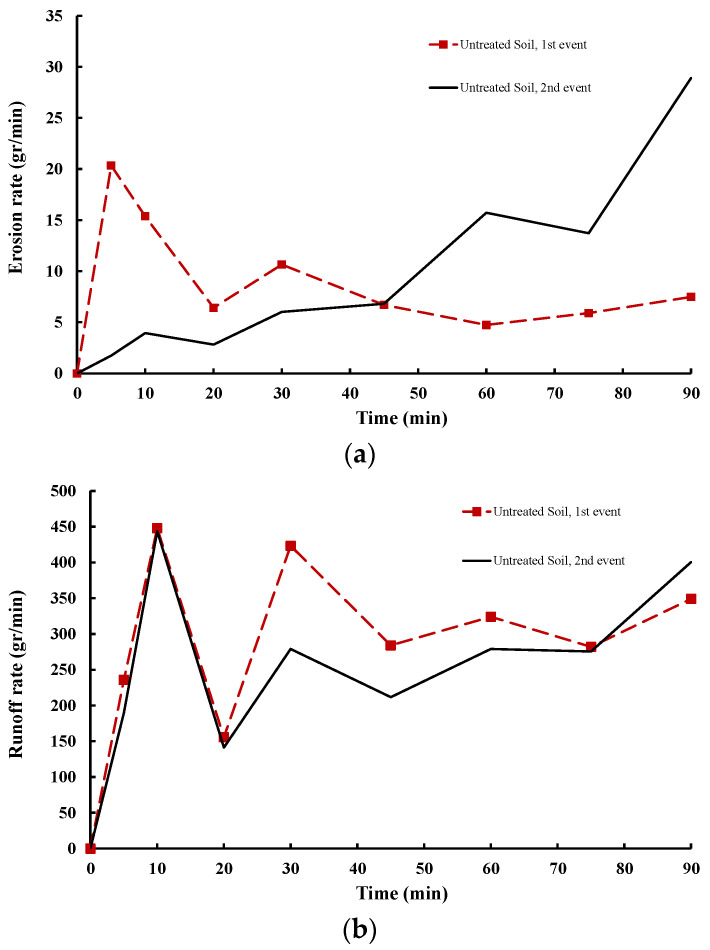
(**a**) Erosion rate and (**b**) rate of surface runoff for untreated soil, 1st and 2nd event.

**Figure 10 polymers-16-01300-f010:**
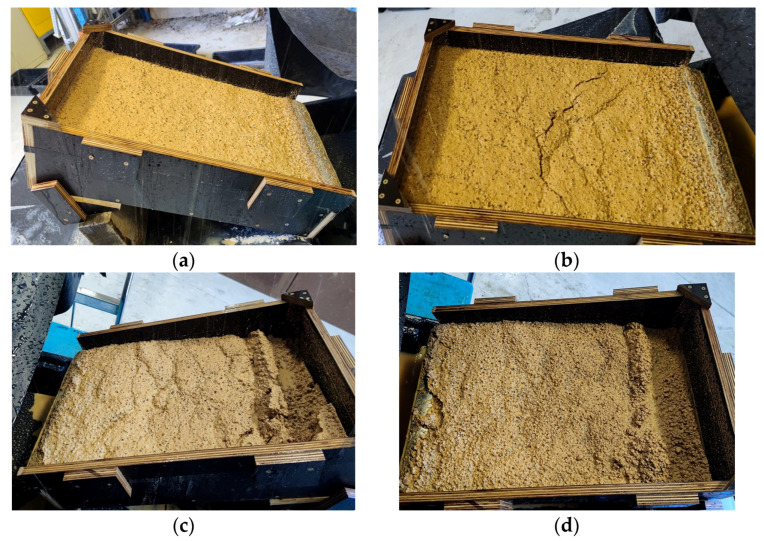
Progress in rainfall-induced flume test, untreated soil, 1st event after (**a**) 1 min, (**b**) 20 min, (**c**) 45 min, and (**d**) 90 min.

**Figure 11 polymers-16-01300-f011:**
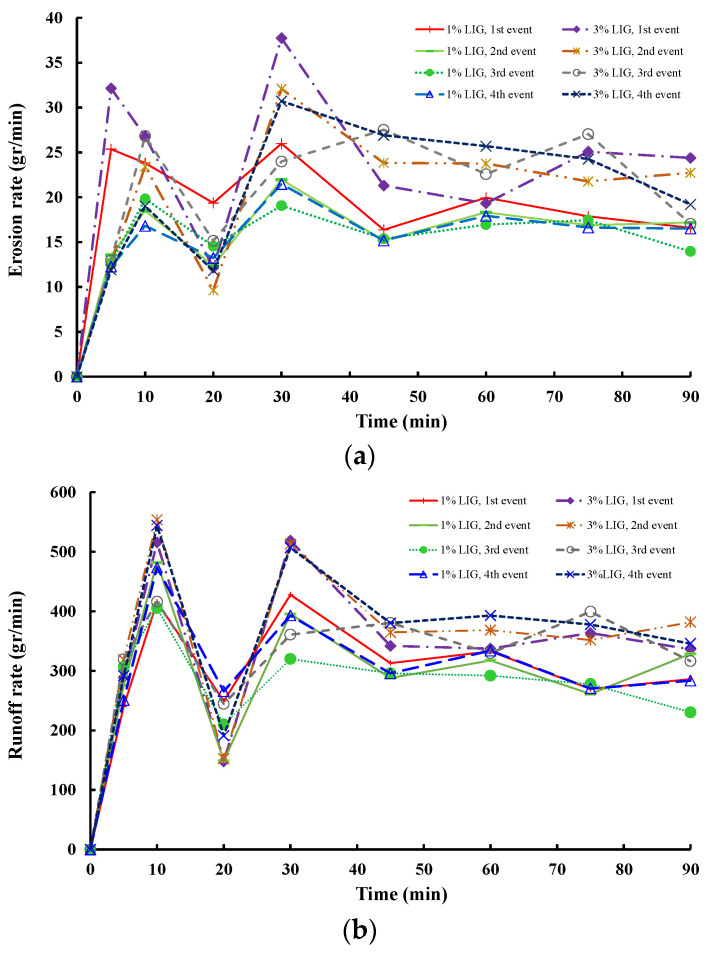
(**a**) Erosion rate and (**b**) rate of surface runoff, for 1% and 3% LIG-treated soil.

**Figure 12 polymers-16-01300-f012:**
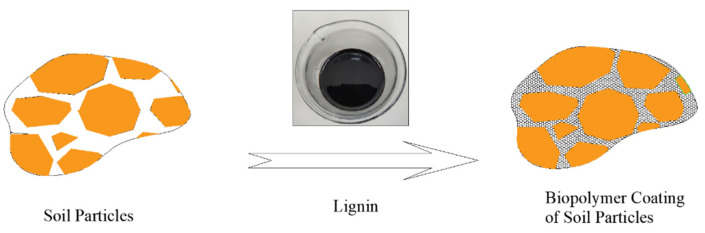
Schematic interaction of LIG biopolymer with soil particles.

**Table 1 polymers-16-01300-t001:** Soil properties.

Soil Type	LL (%)	PL (%)	PI (%)	Specific Gravity
ML	38.0	26.9	11.1	2.77

**Table 2 polymers-16-01300-t002:** Shear strength parameters of untreated and LIG-treated soil.

Soil Reference	Cohesion, c (kPa)	Friction Angle, ϕ (°)
Untreated soil	280	41.0
1% LIG	600	37.6
3% LIG	630	44.4

## Data Availability

The original contributions presented in the study are included in the article, further inquiries can be directed to the corresponding authors.

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
