# Peer review of "Mitigation of Soil Erosion and Enhancement of Slope Stability through the Utilization of Lignin Biopolymer"

_polymers, 2024, doi:10.3390/polym16091300_

Round 1

Reviewer 1 Report

Comments and Suggestions for Authors

Authors used lignin as an additive for soil stabilization. Overall performance is significantly enhanced by adding 3% of lignin. However, there are several points that are not addressed, such as why only 1% and 3% concentration was chosen. Why not higher or lower concentrations tested ?. Overall write up and flow is  good, therefore I recommend accepting the paper after following changes.

  [1].            In abstract “Authors mentioned treated and untreated samples”. Also add some details about how much percentage treated samples … and what type of samples were fabricated should also be clarified. For example “compacted cylinders of soil with ___% of lignin”

  [2].            In the introduction part, the “mechanism of soil reinforcement using biopolymers” is missing. Authors stated that Xenthane, agar, Chitosan etc are tested to be good, however underlying mechanism of such materials is missing.

  [3].            Lignin has poor stability in slightly acidic conditions. Similarly, the basic conditions cause swelling in lignin. Authors should also test the stability of optimized sample in slightly acidic and slightly basic conditions of water.

  [4].            Can author provide the value of swelling of optimized and reference sample.

  [5].            Water contact angle measurements can provide an approximation of modified soil interface with water. By virtue of which, we can judge the interfacial tension difference between both untreated and treated sample.

  [6].            Authors used different sample codes in the manuscript, which cause confusion to the readers. For example, untreated sample and pure soil. Authors should unify all the codes.

  [7].            Figures of run off rate and erosion rate can be merged to reduce the total number of figures. In current form it looks like list of figures, not research paper.

  [8].            Why did authors only chose 1% and 3% lignin addition. Why not higher or lower concentration tested. Please justify chosen concentrations in the introduction section.

  [9].            Underlying mechanism of enhanced performance should be given at the end of this work and schematic diagram highlighting the enhanced performance should also be added.

Optional comment:

If authors can add FTIR or XRD or SEM of samples, it will really increase the quality of paper. Therefore, if accessible in given, please add any of these characterizations.

Editing errors

Line 158 …… 600mm should be 600 mm

Line 159 …… 400mm should be 400 mm

Line 205, remove extra enter .. 206 line should be connected to line 205.

Line 226 …… Cho chang et al conducted .... (reference) is correct way of citing litrature. Similarly, all others should also be corrected. (reference) conducted … is not correct format

Author Response

We would like to thank the reviewer for his/her insightful and helpful comments and suggestions. Based on the advice received, we have improved/modified the manuscript to address questions and concerns.

Response to the Reviewer

Q1: In abstract “Authors mentioned treated and untreated samples”. Also add some details about how much percentage treated samples … and what type of samples were fabricated should also be clarified. For example “compacted cylinders of soil with ___% of lignin”.

The manuscript has been carefully reviewed and revised according to the reviewer’s comments and suggestions. All modifications have been done with track changes throughout the manuscript. The given article was cited in the revised manuscript accordingly.

Q2: In the introduction part, the “mechanism of soil reinforcement using biopolymers” is missing. Authors stated that Xenthane, agar, Chitosan etc are tested to be good, however underlying mechanism of such materials is missing.

The manuscript has been carefully reviewed and revised according to the reviewer’s comments and suggestions. All modifications have been done with track changes throughout the manuscript.

Q3: Lignin has poor stability in slightly acidic conditions. Similarly, the basic conditions cause swelling in lignin. Authors should also test the stability of optimized sample in slightly acidic and slightly basic conditions of water.

We appreciate the reviewer's valuable input regarding the stability of lignin under acidic and basic conditions, which could be crucial for future studies exploring soil stabilization. Although this assessment lies beyond the scope of our current research.

Q4: Can author provide the value of swelling of optimized and reference sample.

The manuscript has been revised according to the reviewer’s comments and suggestions. All modifications have been done with track changes throughout the manuscript.

Q5: Water contact angle measurements can provide an approximation of modified soil interface with water. By virtue of which, we can judge the interfacial tension difference between both untreated and treated sample.

We thank the reviewer for his/her insightful suggestion. While water contact angle measurements could indeed offer valuable insights into the interaction between soil and biopolymer additives, our study primarily focuses on investigating the mechanical properties of soil treated with lignin biopolymer. Regrettably, conducting experiments for water contact angle measurements exceeds the scope of our current research. However, we acknowledge the significance of this aspect for a comprehensive understanding of soil behavior and will consider it for future studies.

Q6: Authors used different sample codes in the manuscript, which cause confusion to the readers. For example, untreated sample and pure soil. Authors should unify all the codes.

The manuscript has been revised according to the reviewer’s comments and suggestions. All modifications have been done with track changes throughout the manuscript.

Q7: Figures of run off rate and erosion rate can be merged to reduce the total number of figures. In current form it looks like list of figures, not research paper.

The manuscript has been revised according to the reviewer’s comments and suggestions. All modifications have been done with track changes throughout the manuscript.

Q8: Why did authors only chose 1% and 3% lignin addition. Why not higher or lower concentration tested. Please justify chosen concentrations in the introduction section.

We appreciate the reviewer's inquiry regarding the selection of 1% and 3% lignin concentrations in our study. Our decision was informed by preliminary testing, which revealed that higher concentrations of lignin (above 3%) presented challenges in terms of workability during mixing with soil, thus limiting their feasibility for our experiments. Additionally, our focus on conducting a series of both laboratory and flume experiments necessitated a practical approach to concentration selection. We observed that very low concentrations of lignin may not yield significant effectiveness compared to higher dosages. Hence, to balance workability and efficacy, we opted to examine the effects of 1% and 3% lignin concentrations. We will ensure to provide a clearer justification for our chosen concentrations in the introduction section of the paper.

Q9: Underlying mechanism of enhanced performance should be given at the end of this work and schematic diagram highlighting the enhanced performance should also be added.

The manuscript has been revised according to the reviewer’s comments and suggestions. All modifications have been done with track changes throughout the manuscript.

Reviewer 2 Report

Comments and Suggestions for Authors

In the manuscript titled"Mitigation of Soil Erosion and Enhancement of Slope Stability through the Utilization of Lignin Biopolymer": This study aimed to investigate the effect of LIG biopolymer against soil erosion and restraining slope failure and enhancement of soil strength by conducting laboratory tests (UU, UCS. Soaking) and fume experiments under uniform rainfall events. The results show that LIG-treated samples demonstrated an enhanced resistance against surface erosion,prevention of slope failure as well as improved shear stress, cohesion, stiffness, and resistance to water infiltration. Here are my peer-review comments:

1. page 3,line 7 .In the 'Materials and Methods' section, please provide more detailed information on any potential variability in the material that could influence reproducibility.

2. The discussion of results could be enhanced by comparing the outcomes with relevant literature, particularly focusing on how LIG treatment compares to other biopolymer treatments in terms of effectiveness and cost.

3. The 'Results' section presents the data clearly, yet the manuscript would benefit from a discussion on the limitations of the laboratory conditions compared to field conditions.

4. Technical terms and acronyms like UCS is used throughout the document; make sure they are defined upon first use in the manuscript.

Author Response

We would like to thank the reviewer for his/her insightful and helpful comments and suggestions. Based on the advice received, we have improved/modified the manuscript to address questions and concerns.

Response to the Reviewer

Q1: 1. page 3, line 7 .In the 'Materials and Methods' section, please provide more detailed information on any potential variability in the material that could influence reproducibility.

Thank you for your valuable feedback. We acknowledge the importance of detailing potential material variability to ensure reproducibility. The ammonium lignosulfonate utilized in our study is a well-known commercialized material sourced from Dustex in Australia. However, it's crucial to note that results may vary if alternative materials are employed.

Q2: The discussion of results could be enhanced by comparing the outcomes with relevant literature, particularly focusing on how LIG treatment compares to other biopolymer treatments in terms of effectiveness and cost.

Thank you for your insightful comment. While comparing our results with relevant literature could indeed provide valuable context, it's essential to note that such comparisons may not always be conclusive or rational. The effectiveness and cost of LIG treatment can vary significantly depending on several factors, including the specific materials used, soil characteristics, and the design of experiments. These variations make direct comparisons challenging and potentially misleading.

Q3: The 'Results' section presents the data clearly, yet the manuscript would benefit from a discussion on the limitations of the laboratory conditions compared to field conditions.

We appreciate the reviewer's insightful comment regarding the need to address the limitations of laboratory conditions compared to field conditions in our study. While our 'Results' section presents clear data demonstrating the effectiveness of lignin-treated soil in mitigating erosion and preventing slope failure, it is essential to acknowledge the inherent constraints of laboratory experiments. Laboratory settings offer controlled environments conducive to precise measurement and evaluation of specific variables. However, they may not fully replicate the complexities of real-world field conditions. Therefore, despite the promising results obtained in our study, it is imperative to recognize the necessity of validating these findings through large-scale field experiments.

The limitations of laboratory experiments have been added to the discussion section.

Q4: Technical terms and acronyms like UCS is used throughout the document; make sure they are defined upon first use in the manuscript.

The manuscript has been revised according to the reviewer’s comment and suggestion. All modifications have been done with track changes throughout the manuscript.

Round 2

Reviewer 1 Report

Comments and Suggestions for Authors

changes are incorporated.